# Leukoencephalopathy with Calcifications and Cysts—The First Polish Patient with Labrune Syndrome

**DOI:** 10.3390/brainsci10110869

**Published:** 2020-11-18

**Authors:** Magdalena Machnikowska-Sokołowska, Jacek Pilch, Justyna Paprocka, Małgorzata Rydzanicz, Agnieszka Pollak, Joanna Kosińska, Piotr Gasperowicz, Katarzyna Gruszczyńska, Ewa Emich-Widera, Rafał Płoski

**Affiliations:** 1Department of Diagnostic Imaging, Radiology and Nuclear Medicine, Faculty of Medical Science in Katowice, Medical University of Silesia, 40-752 Katowice, Poland; magdams@onet.pl (M.M.-S.); kgruszczynska@poczta.onet.pl (K.G.); 2Department of Pediatric Neurology, Faculty of Medical Science in Katowice, Medical University of Silesia, 40-752 Katowice, Poland; japilch@poczta.onet.pl (J.P.); marekwidera@wp.pl (E.E.-W.); 3Department of Medical Genetics, Medical University of Warsaw, 02-106 Warsaw, Poland; mrydzanicz@wum.edu.pl (M.R.); apollak@wum.edu.pl (A.P.); joanna.kosinska@wum.edu.pl (J.K.); piotr.gasperowicz@wum.edu.pl (P.G.); rploski@wum.edu.pl (R.P.)

**Keywords:** Labrune syndrome, *SNORD118* gene, magnetic resonance imaging

## Abstract

Leukoencephalopathy with calcifications and cysts (LCC) is a triad of neuroradiological symptoms characteristic of Labrune syndrome, which was first described in 1996. For 20 years, the diagnosis was only based on clinical, neuroradiological and histopathological findings. Differential diagnosis included a wide spectrum of diseases. Finally, in 2016, genetic mutation in the *SNORD118* gene was confirmed to cause Labrune syndrome. The authors describe a case of a teenage girl with progressive headaches, without developmental delay, presenting with calcifications and white matter abnormality in neuroimaging. Follow-up studies showed the progression of leukoencephalopathy and cyst formation. The first symptoms and initial imaging results posed diagnostic challenges. The final diagnosis was established based on genetic results. The authors discuss the possible therapy of LCC with Bevacizumab.

## 1. Introduction

Progressive leukoencephalopathy with calcifications and cysts (LCC) is a very rare disease with autosomal recessive inheritance, which was first reported by Labrune in 1996 [1]. Since then, only individual cases have been reported. The underlying pathology is a diffuse cerebral microangiopathy with the development of micro- and macrocysts, tumor-like vascular hyperplasia, calcifications, glial proliferation, demyelination, necrosis, iron deposition and hemorrhage. In 2016, it was confirmed that mutations in the *SNORD118* gene were associated with Labrune syndrome. The gene encodes a small nuclear RNA, which is necessary for ribosomal mRNA processing [2].

The challenge in establishing the diagnosis of Labrune syndrome is due to the rare occurrence of the disease, inconsistent clinico-radiological findings and the necessity for follow-up neuroimaging due to the progression of imaging changes on MRI.

## 2. Case Report

The authors report on a normally developing 10-year-old girl who complained of chronic headaches and frequent upper respiratory tract infections without any additional clinical symptoms. The family history was negative. Pregnancy and the delivery period were unremarkable. The 3-year follow-up started with the head CT (at a different hospital) due to recurrent and severe headaches. Non-specific, bilateral, symmetrical, multifocal calcifications in deep brain structures were reported. (Figure 1). After six months, an MRI of the brain was performed. Susceptibility-weighted imaging (SWI) artifacts corresponding to the calcifications on CT and probably small areas of hemorrhage with the areas of periventricular white matter leukodystrophy were found. Additionally, chronic sinusitis was diagnosed. The following were suspected: ischemic-hypoxic changes; post-infectious changes; or Aicardi-Goutières syndrome. Pediatric and neurological examination revealed no abnormalities. Broad neurological assessment included the analysis of the cerebrospinal fluid, which was normal. EEG recording showed epileptiform abnormalities, prominent over the temporal regions, without seizures. Disorders of calcium and phosphate metabolism and other endocrine abnormalities were excluded. Most inborn errors of metabolism were also ruled out during biochemical assessment. After the diagnosis and treatment of pollen allergy, headaches resolved. The authors obtained signed informed consent of the patient’s parents for the publication.

The second brain MRI, which was performed seven months later, showed progression of white matter hyperintensity and the occurrence of a small cyst in the left thalamus (Figure 2). The girl’s psychomotor development was still normal. Considering the progression of radiological changes in the brain, the girl was qualified for genetic testing using whole exome sequencing (WES) with the suspicion of an extremely rare or unknown inborn error of metabolism.

On the third follow-up MRI, which was performed one year later, cysts grew up to 2 cm, showing the mass effect on the left lateral ventricle and white matter hyperintensity with the sparing of the subcortical U-fibers (Figure 3). The suspicion of progressive leukoencephalopathy with calcifications and cysts (LCC) was raised. At that time, pediatric and neurological examination was normal.

## 3. Genetic Analysis

DNA from the proband and her parents was extracted from peripheral blood using the standard protocol. Library preparation for WES was performed using the DNA sample of the proband with the SureSelectXT Human kit All Exon v7 (Agilent, Agilent Technologies, Santa Clara, CA, USA) and paired-end sequence (2 × 100 bp) on HiSeq 1500 (Illumina, San Diego, CA, USA) to obtain 64,810,369 reads (89.4% of target bases were covered at a minimum of 20×, whereas 95.2% had coverage of min. 10×). The obtained raw data were processed with the pipeline previously described [3]. WES data were inspected with an Integrative Genomics Viewer (IGV). Two heterozygous variants in the *SNORD118* gene were prioritized for further verification (NR_033294.1:n.19C > G and NR_033294.1:n.* 5C > G). Both disease-causing variants were validated in the proband, and studied in his parents by amplicon deep sequencing (ADS) (Figure 4).

The transmission of the variants was consistent with the autosomal recessive pattern, as revealed by family assessment. The population frequency for the variants: n.19C > G and n.* 5C > G was 0.00003492 and 0.0006978, respectively, in gnomAD (database, v.3, https://gnomad.broadinstitute.org/) and 0.000834725 and 0.000554939 in in-house datasets of >3500 WES of Polish individuals. Both variants were previously described, although these were only single reports [2,4]. They were reported as causative of LCC, although no comprehensive characteristics of patients with n.19C > G variant were raised.

The suspicion of Labrune syndrome was raised due to cyst progression in the degenerated white matter and increasing calcifications in the follow-up studies (Figure 5). Identified variants in the *SNORD118* gene in correlation with radiological changes confirmed this diagnosis.

## 4. Discussion

In 1996, Labrune et al. described a new disorder in three unrelated children. It was a triad of LCC as a separate entity. Diagnosis of LCC was based on clinical and neuroradiological findings with the histological examination in some patients [1,5].

In 2016, it was confirmed that LCC was a novel single-gene disorder due to germ-line biallelic mutation in the box C/D snoRNA [2]. Those authors analyzed the clinical data and the biological samples of 40 patients with LCC over a period of 12 years. In 2016, Iwama et al. identified eight mutations of the same gene in patients from unrelated families [4]. The last information on genetic background was described by Crow et al. [6].

Due to the progressive nature of the disease, follow-up MRI and the clinical observations are of primary importance, as a single examination is not sufficient to establish the diagnosis [7]. Discrepancy between the relative well-being of our patient and abnormal imaging findings poses difficulties in diagnosis. Patients with LCC may complain of focal neurological deficits, progressive extrapyramidal, pyramidal and cerebellar symptoms, headaches and/or seizures. In most, cognitive deficits are also reported. We observed a discrepancy between clinical and radiological data in our patient. The subject was a normally developing girl who sought medical attention for recurrent headaches only.

Differential diagnosis was difficult, especially with slowly occurring changes in the follow-up imaging studies. First, calcifications in the basal ganglia on CT prompted further diagnostic testing. Basal ganglia calcifications on CT (which were found in our patient in the first neuroimaging) are non-specific, and may occur in many diseases as a primary or secondary sign. They may be idiopathic, toxic (carbon monoxide, lead exposure), infectious (TORCH complex, AIDS), metabolic (hypoparathyroidism, pseudohypoparathyroidism, Cockayne syndrome I and II, mitochondrial diseases (MELAS, MERRF), neuroferritinopathy, neurodegeneration with brain iron accumulation—NBIA) and inherited in origin (chromosomal abnormalities: *SLC20A2 8p11.21 PDGFRB 5q32 PDGFB 22q13.1 XPR1 1q25.3 MYORG 9p13.3).*

In our case, the CT finding of basal ganglia calcifications prompted wide chemical and metabolic testing, which resulted in the exclusion of infectious, toxic or metabolic origin.

Fahr disease can reveal bilateral deep nuclei and subcortical calcifications without white matter abnormalities, which were found in our patient. Endocrinopathies (e.g., hypo, hyper-parathyroidism) were relatively easily excluded by laboratory testing. White matter changes found on the first MRI extended the diagnosis to leukodystrophy with calcifications. However, they were not useful in establishing the diagnosis. When infectious, toxic and hormonal factors were excluded, further metabolic investigation was necessary.

Leukoencephalopathy with possible calcifications is known to occur in patients after radio- or chemotherapy, which was not observed in our case. The development of cysts with the progression of leukoencephalopathy on the third follow-up MRI examination added to the differential diagnosis, which concentrated on LCC.

In Aicardi-Goutières syndrome, severe symptoms can already be present in infancy. Our patient was seven years old, and brain cysts were not typical of this syndrome. The symptoms of RNASET-2 deficient encephalopathy can also be present in infancy. However, the triad of leukoencephalopathy, calcifications and cysts is often observed. The older age of our patient and the clinical picture with further genetic testing were crucial for the differential diagnosis [8]. In Cockayne syndrome, systemic changes, which were absent in our patient, are associated with neurological symptoms. Brain atrophy, white matter signal abnormality and calcifications are the most prevalent in neuroimaging studies. Late-onset Krabbe disease presents different neurological signs, including ataxia, weakness and spasticity. Cystic changes with calcifications require the exclusion of parasitic (hydatid, cysticercosis and cryptococcosis) changes. No serologic evidence for parasitic infestation was found. Parasitic diseases do not typically present with white matter abnormalities.

The Coats plus syndrome resembled Labrune syndrome. In neuroimaging, a triad of symptoms is so similar that before genetic explanation and differentiation between these diseases, they were considered the spectrum of the same pathology [9]. Neurological symptoms and pathologic findings, particularly cerebral microangiopathy [10], posed further problems with differentiation. In these cases, additional retinopathy and sometimes systemic pathology is observed. In 2014, Livingstone et al. suggested a different genetic origin of Coats plus syndrome and LCC, based on negative testing for typical Coats plus, which is related to gene mutation [11]. In the Coats plus syndrome, the genetic origin is different, and the disease presents with the *CTC1* gene abnormality [12]. Therefore, genetic testing is crucial for the final diagnosis. Next to clinical findings and neuroimaging studies [13,14,15,16], genetics has significantly contributed to the explanation of the etiology of the syndrome.

Labrune syndrome is a progressive disorder with onset at different ages, but usually in childhood. Clinical severity is highly variable. In many cases, severe developmental delay, progressive neurological and cognitive decline with premature death are reported. Some patients require neurosurgical intervention due to symptoms of increased intracranial pressure. This clinical variability is most likely related to the location of the cysts, and from a clinical point of view, it fails to distinguish the most characteristic symptoms of LCC. There is also no clear correlation between the type of mutation and clinical symptoms. In our patient only an increase in radiological changes was observed, which so far are not accompanied by any clinical symptoms.

To date, there is no causal treatment in the LCC. Most of the clinical symptoms result from cerebral microangiopathy as tumor-like vascular hyperplasia. One possible disease mechanism may be a malfunction in the signaling pathway of vascular endothelial growth factors (VEGF). In 2017, Fay et al. reported first data on treatment with Bevacizumab, a monoclonal antibody anti-VEGF [17]. After one year of therapy for an 18-year-old patient with Bevacizumab and inhibition of VEGA signaling pathway, the authors observed significant improvement in neurological function, and brain MRI demonstrated a marked reduction in cyst volume and white matter lesions. Considering the lack of clinical symptoms and possible causal treatment, we plan to undertake similar Bevacizumab treatment in our patient in the near future.

## 5. Conclusions

Only follow-up MRI examinations in a normally developing child and a correlation with an in-depth genetic analysis prompted to establish the diagnosis of this rare genetic metabolic syndrome. Cooperation between clinicians, radiologists and geneticists was crucial for establishing the final diagnosis, and for personalized therapy.

## Figures and Tables

**Figure 1 brainsci-10-00869-f001:**
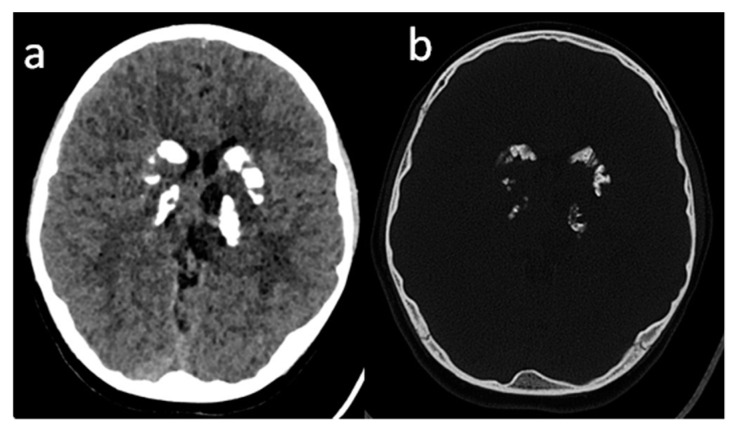
Head CT, axial plane, (**a**) soft tissue window; (**b**) bone window: bilateral, symmetrical and multifocal calcification in deep structures (basal ganglia) and dentate nuclei (not shown).

**Figure 2 brainsci-10-00869-f002:**
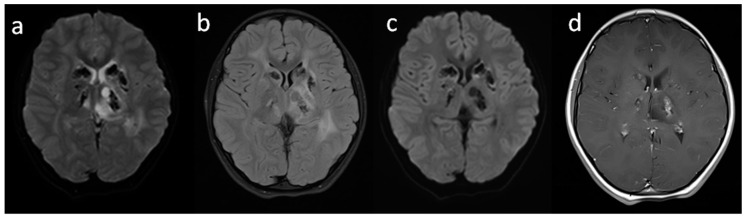
Second brain MRI; one year later, axial plane: (**a**) T2; (**b**) Flair T2; (**c**) T1; (**d**) T1 + CE sequences, corresponding calcification artifacts, small left thalamic cyst and white matter hyperintensity.

**Figure 3 brainsci-10-00869-f003:**
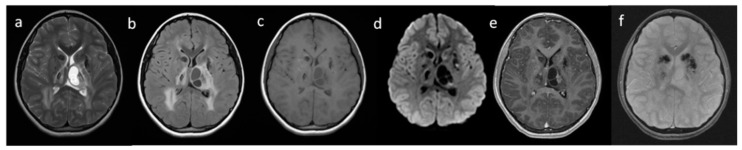
Third brain MRI two years after the first imaging, axial plane: (**a**) T2; (**b**) Flair T2; (**c**) T1; (**d**) DWI; (**e**) T1 + CE; (**f**) SWI (used for the assessment of blood metabolites and calcifications) sequences; white matter hyperintensity progression, cyst enlargement with the mass effect and calcification artifacts.

**Figure 4 brainsci-10-00869-f004:**
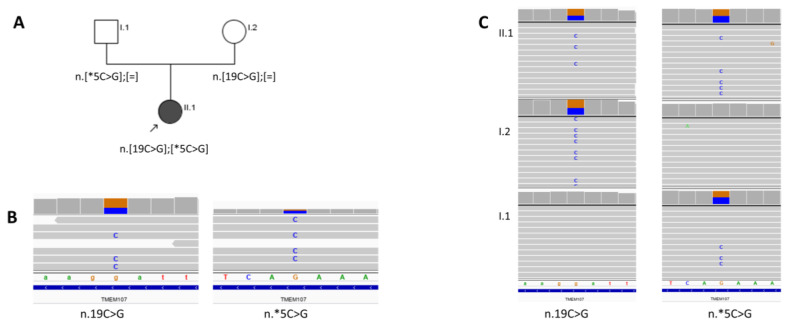
Pedigree of the studied family (**A**) and the results of WES and ADS for the variants in the *SNORD118* gene; (**B**) WES results; (**C**) results of amplicon deep sequencing in the proband, her parents (I–proband, II–mother, III–father). The obtained coverage of the ADS for the n.19C > G position in the *SNORD118* gene was: 9508 reads for the proband (II.1); 8923 reads for the mother (I.2); and 7578 reads for the father (I.1). The obtained coverage of the ADS for the n.* 5C > G position in the *SNORD118* gene was: 11,529 reads for the proband (II.1); 7721 reads for the mother (I.2); and 8440 reads for the father (I.1).

**Figure 5 brainsci-10-00869-f005:**
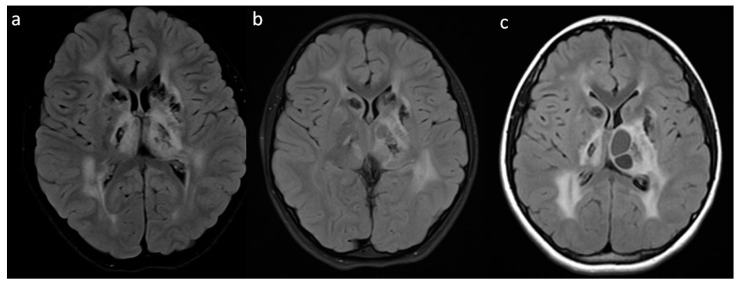
Follow-up MRI of the brain, axial plane in the T2 FLAIR sequence; (**a**) First MRI—at the age of 7 years; (**b**) Second MRI 6 months later; (**c**) Third MRI 1 year later—progression of calcifications, cysts in the left thalamus with the mass effect on MRI and white matter abnormalities.

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
