# Peer review of "Leukoencephalopathy with Calcifications and Cysts—The First Polish Patient with Labrune Syndrome"

_brainsci, 2020, doi:10.3390/brainsci10110869_

Round 1
Reviewer 1 Report
This is an interesting and well-studied case report on a 10-year-old girl with
headache and a brain MRI characterized by calcifications and white matter abnormalities with two mutations of SNORD118 32 gene. The genetic findings allowed to formulate a diagnosis of Labrune syndrome, for the first time in Poland.
In the Abstract, the Authors state that this girl had seizures, but in the following Case Report they describe only headache events, not epileptic seizures. Then, I think that the Authors should delete the word "seizures" in the Abstract and in the text.
In the Introduction (last line) the Authors state: "Additionally, a very small gene may be overlooked in high-throughput genetic testing". I think that this sentence is unhelpful and simplistic, and it should be deleted. In fact, the Authors correctly performed WES in this case, and nobody would carry out any other non appropriate genetic tests, such as karyotype, or array-CGH in a leukoencephalopathy.
In the Case Report section, the Authors do not provide information on the electroencephalogram of this girl. Was EEG recorded? Was it normal? Did it show epileptiform abnormalities, as often is the case in leucoencephalopathies.
The Discussion treats very well the differential diagnosis of Labrune syndrome. However, the Authors should also compare their case with the other ones in literature, in order to discuss similarities or discordancies. A summary table with the main clinical, EEG, and MRI features of the literature cases and the new Polish patient could be adequate.
Author Response
We would like to thank for your valuable comments. During revisions we made some additional corrections. We clearly highlighted revisions using the "Track Changes" function in Microsoft Word.
Below we list the details of the revisions made in response to your comments:
In the Abstract, the Authors state that this girl had seizures, but in the following Case Report they describe only headache events, not epileptic seizures. Then, I think that the Authors should delete the word "seizures" in the Abstract and in the text.
Thank you for this remark. We deleted the word „seizures”.
In the Introduction (last line) the Authors state: "Additionally, a very small gene may be overlooked in high-throughput genetic testing". I think that this sentence is unhelpful and simplistic, and it should be deleted. In fact, the Authors correctly performed WES in this case, and nobody would carry out any other non appropriate genetic tests, such as karyotype, or array-CGH in a leukoencephalopathy.
We deleted the sentence: "Additionally, a very small gene may be overlooked in high-throughput genetic testing".
In the Case Report section, the Authors do not provide information on the electroencephalogram of this girl. Was EEG recorded? Was it normal? Did it show epileptiform abnormalities, as often is the case in leucoencephalopathies.
The EEG was abnormal with epileptiform abnormalities (with temporal predominance) without clinical presentation.
The Discussion treats very well the differential diagnosis of Labrune syndrome. However, the Authors should also compare their case with the other ones in literature, in order to discuss similarities or discordancies. A summary table with the main clinical, EEG, and MRI features of the literature cases and the new Polish patient could be adequate.
We have made the changes according to your suggestions. We added a new paragraph as follows:
This clinical variability is most likely related to the location of the cysts, and from a clinical point of view, it fails to distinguish the most characteristic symptoms of LCC. There is also no clear correlation between the type of mutation and clinical symptoms. In our patient was only observed an progression of radiological changes which, so far, are not accompanied by any clinical symptoms.
Reviewer 2 Report
The authors report the clinical and radiographic features of a case of Labrune syndrome associated with compound heterozygous SNORD118 mutations and describe the differential diagnosis associated with neuroimaging findings.
This is an interesting case report. However, I fail to see how it enriches the literature on SNORD118-related leukoencephalopathy. The discussion only summarizes the differential diagnosis which was mostly implied in the case description. This section should instead highlight why this case report is worth publishing.
Unfortunately, as the case report fails to highlight a specific or new phenotype, a new approach to management or further insight into disease pathophysiology.
Author Response
We would like to thank for your valuable comments. During revisions we made some additional corrections. We clearly highlighted revisions using the "Track Changes" function in Microsoft Word.
Below we list the details of the revisions made in response to your comments
This is an interesting case report. However, I fail to see how it enriches the literature on SNORD118-related leukoencephalopathy. The discussion only summarizes the differential diagnosis which was mostly implied in the case description. This section should instead highlight why this case report is worth publishing.
We explained as follows:
This clinical variability is most likely related to the location of the cysts, and from a clinical point of view, it fails to distinguish the most characteristic symptoms of LCC. There is also no clear correlation between the type of mutation and clinical symptoms. In our patient was only observed an progression of radiological changes which, so far, are not accompanied by any clinical symptoms.
Unfortunately, as the case report fails to highlight a specific or new phenotype, a new approach to management or further insight into disease pathophysiology.
We have made the changes according to your suggestions. We added a new paragraph as follows:
To date, there is no causal treatment in the LCC. Most of the clinical symptoms result from cerebral microangiopathy as tumor-like vascular hyperplasia. One possible disease mechanism may be a malfunction in the signaling pathway of vascular endothelial growth factor (VEGF). In 2017 Fay et al. reported first data on treatment with Bevacizumab, a monoclonal antibody anti-VEGF [17]. After one year therapy of 18-years-old patient with Bevacizumab and inhibition of VEGA signaling pathway the authors observed significant improvement in neurological function and brain MRI demonstrated a marked reduction in cyst volume and white matter lesions. Considering the lack of clinical symptoms and possible causal treatment, we plan to undertake similar Bevacizumab treatment in our patient in the near future.
In Conclusions we added:
Cooperation between clinicians, radiologists, and geneticists was crucial for establishing the final diagnosis and for personalized therapy.
Round 2
Reviewer 1 Report
The Authors took in count all the reviewers' queries, and I think that this paper is now imporved.
Just a last, minor point:
At page 2, line 65 the Authors added "EEG recording showed epileptiform abnormalities without seizures". I think that they should write: "EEG recording showed epileptiform abnormalities, prominent over the temporal regions, without seizures"
Author Response
Dear Reviewer,
Thank you for your comment. We corrected the sentence concerning EEG.
The Authors
Reviewer 2 Report
Comments adequately addressed.
Author Response
Dear Reviewer,
Thank you for your comments and suggestions during review.
The Authors